# Evolutionary Adaptations in Biliverdin Reductase B: Insights into Coenzyme Dynamics and Catalytic Efficiency

**DOI:** 10.3390/ijms252413233

**Published:** 2024-12-10

**Authors:** Eunjeong Lee, Jasmina S. Redzic, Elan Zohar Eisenmesser

**Affiliations:** Department of Biochemistry and Molecular Genetics, School of Medicine, University of Colorado Anschutz Medical Campus, Aurora, CO 80045, USA; eunjeong.lee@cuanschutz.edu (E.L.); jasmina.redzic@cuanschutz.edu (J.S.R.)

**Keywords:** NMR, BLVRB, protein dynamics, catalysis

## Abstract

Biliverdin reductase B (BLVRB) is a redox regulator that catalyzes nicotinamide adenine dinucleotide phosphate (NADPH)-dependent reductions of multiple substrates, including flavins and biliverdin-β. BLVRB has emerging roles in redox regulation and post-translational modifications, highlighting its importance in various physiological contexts. In this study, we explore the structural and functional differences between human BLVRB and its hyrax homologue, focusing on evolutionary adaptations at the active site and allosteric regions. Using NMR spectroscopy, we compared coenzyme binding, catalytic turnover, and dynamic behavior between the two homologues. Despite lacking the arginine “clamp” present in human BLVRB, hyrax BLVRB still undergoes conformational changes in response to the oxidative state of the coenzyme. Mutations at the allosteric site (position 164) show that threonine at this position enhances coenzyme discrimination and allosteric coupling in human BLVRB, while hyrax BLVRB does not display the same allosteric effects. Relaxation experiments revealed distinct dynamic behaviors in hyrax BLVRB, with increased flexibility in its holo form due to the absence of the clamp. Our findings suggest that the evolutionary loss of the active site clamp and modifications at position 164 in hyrax BLVRB alter the enzyme’s conformational dynamics and coenzyme interactions. Identified similarities and differences underscore how key regions modulate catalytic efficiency and suggest that coenzyme isomerization may represent the rate-limiting step in both homologues.

## 1. Introduction

Biliverdin reductase B (BLVRB) is a ubiquitously expressed redox regulator that catalyzes the NADPH-dependent reduction of multiple substrates. Unlike the BLVRA isoform that solely catalyzes the reduction of biliverdin-α to bilirubin-α, BLVRB catalyzes the reduction of biliverdin-β (to bilirubin-β) and acts as a general flavin reductase that reduces both flavin adenine dinucleotide (FAD) and flavin mononucleotide (FMN) [1]. Recently, BLVRB has also been identified as an S-nitrosylase, mediating the post-translational modification of various proteins [2]. This broad substrate specificity suggests BLVRB plays a central role in redox chemistry, consistent with its high expression across multiple cell types, including red blood cells [3]. Notably, BLVRB is critical for hematopoietic cell fate, determining monocyte development or, in its absence, platelet production [4], making it a potential target for increasing platelet levels in thrombocytopenia [5]. Additionally, targeting BLVRB activity in cancer cells could disrupt their metabolism and sensitize them to metabolic inhibitors [6], while in other contexts increasing BLVRB expression may suppress the epithelial–mesenchymal transition, offering a strategy to prevent metastasis [7]. These findings underscore the importance of fully characterizing BLVRB, as both inhibition and activation of its activity hold therapeutic potential.

Because of its clinical relevance, we have spent considerable effort in elucidating the underlying steps that dictate BLVRB function that include searching for the rate-limiting step of substrate turnover. Through our studies, we have revealed a uniquely evolving enzyme active site that modulates function. Specifically, our studies began by identifying large conformational changes to BLVRB due to coenzyme (NADP^+^) interactions that were associated with a sub-micromolar affinity, which were in contrast to the relatively weak near-millimolar affinity interactions with flavin substrates [3]. Such studies identified an evolutionarily changing active site “clamp” at either position 14 or position 78 that straddle the coenzyme (Figure 1A,B), with some homologues comprising two arginine clamps at both positions and some comprising no such arginine clamp. For example, X-ray crystallography had previously revealed that human BLVRB with a single arginine at one of these positions (Q14, R78) forms one clamp [8]. Later, our own crystallographic analysis revealed that a lemur BLVRB homologue with an arginine at the other active site position also forms a clamp (R14, G78), while a human BLVRB Q14R mutant that emulates other mammalian homologues (R14, R78) comprises two clamps (Figure 1A,C) [9]. NMR solution studies revealed that coenzyme release was inversely related to the number of clamps, and, thus, the function of each clamp is to systematically slow the oxidized NADP^+^ product release. For example, the human BLVRB active site mutant of Q14R that comprises two arginine clamps exhibits the slowest coenzyme off-rate, while hyrax BLVRB that comprises no active site clamp exhibits the fastest off-rate [9]. Surprisingly, slower coenzyme release was associated with an increased catalytic turnover, suggesting such steps could be coupled but that coenzyme release is not rate-limiting for BLVRB. Considering that all BLVRBs and mutations exhibit hydride transfer rates of 30–40 s^−1^, coenzyme products release off-rates of 1–15 s^−1^, but catalytic turnover is only ~0.1 s^−1^, there remains an unidentified rate-limiting step that occurs after hydride transfer from NADPH to NADP^+^ [9].

Recent studies suggest the rate-limiting step may correspond to coenzyme isomerization after hydride transfer. Large chemical shift perturbations (CSPs) between NADPH and NADP^+^-bound forms of human BLVRB indicate a conformational exchange consistent with coenzyme isomerization [10]. Similar oxidative-state-dependent changes have been observed in other enzymes. For example, in *Rhodospirillum rubrum* transhydrogenase, NMR studies reveal large spectral shifts related to coenzyme redox states [11], while X-ray crystallography of ferredoxin-bound coenzyme demonstrated redox-state-dependent changes in stereochemistry [12]. Despite such observations, crystal structures often show similar conformations for both oxidative states, likely due to constraints imposed by crystallization conditions [11,13]. This highlights NMR as a particularly effective method for studying conformational changes that result from coenzyme isomerization in solution. Here, we aim to determine whether similar global CSPs occur in other BLVRB homologues, focusing on hyrax BLVRB, which lacks an active site arginine clamp.

In addition to a uniquely evolving BLVRB active site, we have also identified an allosteric site at position 164 that evolutionarily alternates between threonine and serine across species (Figure 1A,D). For example, in human BLVRB, T164 is inherently dynamic on a slow micro-millisecond timescale. However, mutating T164 to serine quenched this dynamic behavior that significantly altered coenzyme release and was also coupled to the slower unidentified rate-limiting step [14]. This suggests that the 164 site has evolved in humans to play an important role in allosteric regulation at multiple stages of function. Whether such a distal allosteric site is also coupled to coenzyme release and a rate-limiting step in different BLVRB homologues also comprises one of our goals here.

To further elucidate both active site and distally coupled evolutionary differences and to determine whether a potential coenzyme isomerization occurs within multiple homologues, we expand BLVRB studies here by assigning the hyrax BLVRB homologue for NMR studies. Hyrax BLVRB shares 89% identity with human BLVRB (Figure 1B) but exhibits key variations that we show here are coupled to functional differences. Unlike human BLVRB, hyrax BLVRB does not comprise an active site arginine and, thus, does not comprise an active site clamp. Hyrax BLVRB also exhibits nearly complete resonance coverage, in contrast to other homologues that are missing several NMR resonances due to inherent intermediate timescale dynamics. This allows us to explore how structural features, such as the absence of an arginine clamp and the presence of serine at position 164, contribute to differences in coenzyme binding and catalytic turnover. By comparing human and hyrax BLVRB homologues, we aim to better understand how allosteric regulation and active site variability influence BLVRB function. Using NMR, we probed the structural and dynamic differences between these homologues. We find in hyrax BLVRB that active site conformational changes mediated by the oxidative state of the coenzyme occur, as previously identified in human BLVRB [15]. However, there are specific differences in the induced changes between the two homologues. This suggests that both BLVRB homologues undergo isomerization after hydride transfer that likely represents the rate-limiting step. Moreover, by probing swapped mutants, human BLVRB T164S and hyrax BLVRB S164T, we show that this distal site is coupled to coenzyme binding for both homologues but does not affect the rate-limiting step in hyrax BLVRB. Such studies suggest both a BLVRB active site clamp and allosteric coupling to position 164 have evolved in many species to increase the catalytic rate.

## 2. Results

### 2.1. Hyrax BLVRB Is Amenable to NMR Solution Studies

We extended our NMR investigations from previous studies that have compared human and lemur BLVRBs, which exhibit 16 changes between them, to new homologues with varied outcomes. Specifically, we were able to purify three new BLVRB homologues to obtain NMR spectra, which included hyrax BLVRB (ENSPCAT00000003866.1), *Aedes aegypti* (XP_001649677.1), and *Staphylococcus aureus* BLVRB (WP_037589155). These homologues share 89%, 50%, and 27% identity with human BLVRB, respectively. While both hyrax BLVRB and *Staphylococcus aureus* BLVRB gave rise to well-dispersed spectra, both in the absence and presence of the NADP^+^ coenzyme (Appendix A), this was not the case for *Aedies aegypti* BLVRB. Instead, *Aedes aegypti* BLVRB gave rise to a poorly dispersed spectrum in its apo state but gave rise to good dispersion upon NADP^+^ binding (Appendix A). This suggests that *Aedes aegypti* BLVRB may partially unfold in its apo form, resembling a molten globule state, but becomes well folded in its holo form. While this dependence of *Aedes aegypti* BLVRB on coenzyme for folding is intriguing, the complex was unstable over the course of hours at room temperature precluding detailed studies. *Staphylococcus aureus* BLVRB was also unstable over the course of hours at room temperature. Thus, considering that hyrax BLVRB was stable for days at room temperature in both its apo and holo forms, we focused our NMR studies on hyrax BLVRB.

Hyrax BLVRB comprises 23 sequence differences to human BLVRB with several key differences that distinguish this homologue from other family members. For example, unlike most BLVRBs that each comprise an active site arginine clamp at either position 78 or 14, respectively, hyrax BLVRB does not comprise any such arginine (Figure 1C). Instead, hyrax BLVRB comprises M14 and G78 and is therefore unique in that it cannot form a stabilizing clamp over the coenzyme. We have previously shown that the absence of an active site clamp results in a faster off-rate, with weaker coenzyme affinity, that includes hyrax BLVRB [9]. Furthermore, hyrax BLVRB comprises a serine at position S164, similar to lemur S164 but differing from human BLVRB T164. Thus, the mutation of hyrax BLVRB S164 would allow us to determine whether this site relays allosteric regulation to its active site, as observed for mutations of human BLVRB S164 [14]. Finally, hyrax BLVRB residues 166–174 are observed, as opposed to other homologues where their dynamic timescale of exchange likely results in their absence at multiple temperatures. The fact that this region is adjacent to the active site now allows us to utilize more nuclear probes to determine the relationship between structure, dynamics, and function for the entire C-terminal lobe within hyrax BLVRB.

### 2.2. Hyrax BLVRB Reorganization from Apo to Holo Forms Are Distinct from Human BLVRB

Protein backbone Cα resonances are sensitive measures to secondary structure [16], and, thus, their changes (ΔCα) were used here as a proxy to monitor structural changes for hyrax BLVRB upon NADP^+^ binding (Figure 2B). These changes indicate largescale reorganization of the active sites for hyrax BLVRB, which is also observed in human BLVRB, as previously reported [3,14]. However, there are both similarities and distinct differences between the human and hyrax BLVRB homologues observed in solution when comparing their responses to NADP^+^ binding (Figure 2).

Both hyrax and human BLVRB exhibit both environmental and secondary structure differences upon NADP^+^ binding within their active sites (Figure 2A,B). This means that despite the absence of an arginine clamp within hyrax BLVRB (at either position 14 or 78), there remains a reorganization. In fact, the absence of the active site clamp in hyrax BLVRB facilitates larger secondary structural changes for residues 83–84 than human BLVRB that form the substrate binding site adjacent to the coenzyme. This suggests that the increased flexibility of the active site position of hyrax BLVRB G78 may allow for larger changes. Conversely, there are several regions that are not as impacted in hyrax BLVRB by NADP^+^ binding as they are in human BLVRB. For example, while residues near the adenosine moiety, residues 34–40, and more distant regions within the C-terminal lobe exhibit differences between the apo and coenzyme bound in both homologues, the imparted changes upon coenzyme binding are not identical. Specifically, hyrax BLVRB does not undergo as large of a change upon coenzyme binding as that observed in human BLVRB. Thus, in the absence of the clamp in hyrax BLVRB, the coenzyme-induced changes are not as large as they are in the presence of the clamp within human BLVRB, but there are distinct changes upon coenzyme binding for each homologue (Figure 2C,D).

### 2.3. Allosteric Regulation of Coenzyme Binding Affinity in Human and Hyrax BLVRB: The Role of Residue 164

To further investigate how NADPH and NADP^+^ interact with human and hyrax BLVRB, we introduced mutations at position 164, a known allosteric regulator located 16 Å from the active site [14]. We previously mutated human BLVRB’s threonine at position 164 (T164) to serine (S164, as in hyrax) [14]. Here, we mutated hyrax BLVRB’s serine (S164) to threonine (T164, as in human). These mutations allow us to assess how the presence of threonine or serine at this key allosteric site affects coenzyme binding in both BLVRB homologues.

We used isothermal titration calorimetry (ITC) to measure the binding affinities of both wild-type (WT) and mutant forms of BLVRB to NADPH and NADP^+^. All four forms—human WT, human T164S, hyrax WT, and hyrax S164T—showed similar affinities for NADPH, with reduced affinities for NADP^+^ (Figure 2E, Table 1, Appendix A). Importantly, in both human BLVRB WT and hyrax BLVRB S164T, where threonine is present at position 164, NADP^+^ bound weaker than in the context of a serine at this position, such as human BLVRB T164S and hyrax BLVRB WT. This suggests that for both BLVRB homologues, the presence of threonine at position 164 enhances the enzyme’s ability to differentiate between NADPH and NADP^+^.

### 2.4. The Oxidative State of the Coenzyme Induces Largescale Changes for Both BLVRB Homologues That Suggests Coenzyme Isomerization

We recently identified largescale structural changes in human BLVRB between the oxidative states of the coenzyme [10], indicating that hydride transfer to the substrate likely leads to coenzyme conformational rearrangements prior to coenzyme release (i.e., isomerization of the bound coenzyme). Identifying whether this largescale conformational change occurs in multiple homologues here would be especially important, considering that we previously showed that coenzyme release is not rate-limiting [9]. This prompted us to investigate whether this difference in coenzyme oxidative states occurs across homologues, especially the hyrax BLVRB that does not comprise an active site clamp, as it could suggest that coenzyme isomerization occurs and could therefore be the rate-limiting step.

To stabilize NADPH for these studies, we employed the same glucose-6-phosphate dehydrogenase recycling system to hyrax BLVRB that was previously applied to human BLVRB. While both homologues displayed hydride-induced changes adjacent to the active site at residues 152–154, distal changes that extended 15–20 Å from the coenzyme were different (Figure 3A,B). For example, larger CSPs for hyrax BLVRB were observed for residues 77–80. This was likely due to larger secondary structural changes monitored by ΔCα of D80 within hyrax BLVRB. Importantly, our previous mutagenic studies of human BLVRB revealed that removing R78, and thereby removing the active site clamp as in hyrax BLVRB, resulted in lower catalytic turnover similar to the hyrax BLVRB homologue [9]. These studies here indicate that the conformational changes are different between oxidative states of the bound coenzyme between homologues. Nonetheless, both homologues undergo reorganization between bound oxidative states of the coenzyme. Such changes could suggest that such conformational changes reflect differences in the unidentified rate-limiting step, such as coenzyme isomerization after hydride transfer. In fact, we also performed a ^13^C-filtered NOESY experiment, using ^13^C-labeled human BLVRB and unlabeled NADPH and NADP^+^ in attempt to collect enough NOEs to determine their bound structural differences (Appendix A). Although the NOEs were too sparse to determine a high-resolution structure of the bound coenzymes, the resonance positions markedly changed within the ribose ring. This once again indicates a structural reorganization of the active site that is dependent on the oxidative state of the coenzyme and could reflect the rate-limiting step of coenzyme isomerization.

For the allosteric site of residue 164, hyrax BLVRB had a markedly diminished response to the oxidative states. This suggests that, unlike the human BLVRB active site, hyrax BLVRB S164 is not coupled to the active site. To further explore the role of residue 164 in hydride-mediated allosteric coupling, we compared the ^15^N-HSQC spectra of the apo, NADPH-bound, and NADP^+^-bound forms of hyrax and human BLVRB homologues along with their position 164 mutational swaps (Figure 3C,D). Hyrax BLVRB showed no obvious coupling between these forms at position S164, in contrast to that of human BLVRB T164. However, mutating hyrax BLVRB S164 to threonine (S164T) restored partial allosteric coupling, as evidenced by hydride-induced conformational changes at position 164 within hyrax BLVRB (Figure 3C). This suggests that the presence of threonine at this position is critical for hydride-mediated long-range communication between the allosteric site and active site. Conversely, mutating human BLVRB’s T164 to serine partially disrupted the allosteric coupling between the free and NADPH forms (Figure 3D), further supporting the role of this site in mediating species-specific responses to hydride transfer.

These findings highlight the evolutionary divergence in both active site conformational changes and allosteric regulation mechanisms coupled to the oxidative state of the coenzyme but also highlight coenzyme isomerization as a potential rate-limiting step. Such distinctions suggest that the role of the active site clamp and allosteric coupling underlies their catalytic differences.

### 2.5. Substrate and Product Binding Are Similar for Human and Hyrax BLVRB

Our previous BLVRB investigations revealed that the binding affinity for FAD was significantly weaker compared to coenzyme affinity [3,9]. Moreover, mutagenic changes in the clamp within human BLVRB at positions 14 or 78 did not notably affect FAD binding, indicating that substrate binding is largely independent of the clamp once the coenzyme is in place. Here, we extended these substrate-binding studies to include hyrax BLVRB and used NMR for the first time to test the biologically relevant product, bilirubin-β (BR-β). We chose this pair, the FAD substrate and bilirubin-β product, as the FADH_2_ product is highly unstable and bilirubin-β is not commercially available.

As seen in human BLVRB, titration experiments with FAD showed fast-exchange binding in hyrax BLVRB (Figure 4A), allowing us to extract the dissociation constants (i.e., K*d*) from their respective binding isotherms (Figure 4B). In general, both homologues exhibit very similar CSPs for FAD binding (Figure 4C). However, once again, human BLVRB exhibits a larger response than that of hyrax BLVRB for position 164. This suggests that this distal 164 position is once again allosterically coupled to the FAD substrate binding in human BLVRB but not in hyrax BLVRB.

Next, we sought to look at a bound product and chose to investigate the binding of bilirubin-β (BR-β), a physiological product of BLVRB enzymatic activity. Our recent studies of commercial BR preparations had identified two distinct resonances in NMR titrations with human BLVRB [10], which were hypothesized to correspond to the BR-α and BR-β isomers. To test this, we isolated BR-β from a commercial source using HPLC (Appendix A). We then confirmed the identities of the α, β, and γ isomers using 1D NMR (Appendix A). Human BLVRB titrations with BR-β confirmed that one of the bound split resonance groups was indeed specific for this bound isomer that all surrounded the substrate binding site (Appendix A). Furthermore, a titration of human BLVRB with BR-α revealed an expected weak binding affinity (Appendix A). Conversely, titrations of both human and hyrax BLVRB with the specific BR-β isomer exhibited significantly tighter binding affinities, thereby biophysically confirming the isomer-specific binding preference (Figure 4D,E). Notably, some residues also displayed slow exchange. Analogous to the FAD titrations, the induced CSPs for both BLVRB homologues were similar overall, except for position 164 that was once again only coupled in human BLVRB (Figure 4F).

These studies indicate that both hyrax BLVRB and human BLVRB engage their FAD substrate with relatively weak affinity and their BR-β product similarly within uncertainty. Such findings suggest it may be the coenzyme, and not the substrate, that modulates a rate-limiting step.

### 2.6. Position 164 Is Not Coupled to a Rate-Limiting Step in Hyrax BLVRB

Our previous studies have shown that position 164 in human BLVRB is coupled to both coenzyme release and an even slower rate-limiting step [9], prompting us here to determine whether position 164 is also coupled to a rate-limiting step for hyrax BLVRB. This previous study revealed that coenzyme-release (i.e., NADP^+^) was on the order of seconds, yet catalytic turnover is over 10-fold slower, suggesting another conformational step is rate limiting during Michaelis–Menten kinetics. We therefore performed UV kinetics on both human and hyrax BLVRBs, as well as their position 164 swaps (Figure 5, Table 1). The BLVRB UV kinetics assay is particularly challenging, owing to the absorbance of FAD that overlaps with the spectroscopic readout of NADPH depletion at 340 nm [9]. However, we were able to extend this assay to higher FAD concentrations to more accurately quantify the kinetic parameters.

As previously reported, human BLVRB exhibits both a higher k_cat_ than hyrax BLVRB, and the human BLVRB T164S mutant diminishes this turnover rate to that of hyrax BLVRB WT (Figure 5, Table 1). However, the reverse mutation of hyrax BLVRB S164T does not restore its catalytic efficiency to that of its human homologue, indicating that there is a loss of coupling to this site for the rate-limiting step. Thus, as opposed to coenzyme affinity that can be modulated for both human and hyrax BLVRB homologues by mutating position 164, modulation of this position does not alter a rate-limiting step in Michaelis–Menten kinetics for hyrax BLVRB, as is the case for human BLVRB.

### 2.7. Hyrax BLVRB Exhibits Segmental Motions Partially Quenched by Coenzyme Binding

Human BLVRB exhibits both fast (ps-ns) and slow (μs-ms) timescale motions that we previously showed are coupled to coenzyme release despite being much faster [3], and here we sought a similar analysis for the hyrax BLVRB homologue. For hyrax BLVRB, ps-ns motions were probed by R1 relaxation rates for apo and holo forms (Figure 6A). Such R1 relaxation rates were partially diminished in their holo forms, indicating less disorder (i.e., more order in the presence of coenzyme). While this diminishment is obvious for the active site region of residues 77–90, increased R1 relaxation rates, and therefore flexibility, still persists more so in hyrax BLVRB than that of its human homologue that is almost entirely quenched in this fast timescale [3]. For slower μs-ms timescales, we compared the first intensity-based relaxation point derived from R_2_-CPMG dispersion data further described below (Figure 6B). Representative full R_2_-CPMG dispersions are shown for a subset of these (Figure 6C). Again, R_2_ relaxation rates were partially quenched in the holo form relative to the apo form. Despite this dynamic quenching in both BLVRB homologues, these changes for hyrax BLVRB contrasts with that previously revealed for human BLVRB [3]. Specifically, unlike R_2_ relaxation rates for human BLVRB that are completely quenched for residues 72–87 and 109–114, several of these residues still exhibit elevated R_2_ relaxation rates with the coenzyme in hyrax BLVRB. Conversely, coenzyme binding induces exchange in human BLVRB within residues 37–39 adjacent to the adenosine moiety, which is not observed in hyrax BLVRB here. Thus, there are differences in the dynamic response to coenzyme binding between homologues in multiple timescales.

We further interrogated the slower timescale motions through full R_2_-CPMG analysis. R_2_-CPMG dispersions provide the details of μs-ms beyond a single diagnostic of whether motions exist, as the biophysical parameters that underlie the motions can be extracted. These include the rates of motions (*k_ex_*), the populations (of sampled high-energy conformations in a simple two-state model P_A_, P_B_), and the chemical shift change between the two states (Δν_cpmg_), reviewed extensively elsewhere [17]. A simple two-state model was used, as there is no further information to imply more complicated models, and data were collected at two static fields that were both sufficient and necessary for fitting. Although we have previously collected human BLVRB R_2_-CPMG dispersions at 900 MHz at multiple temperatures [10], here we collected 600 MHz for this homologue and both 900 MHz and 600 MHz data for hyrax BLVRB to directly compare their dynamics at 20 °C. This is because at least two static fields are necessary to extract the underlying biophysical parameters accurately [18].

Using R_2_-CPMG dispersions fit simultaneously at both 600 and 900 MHz (Figure 7A,B and Appendix A), human and hyrax BLVRB homologues are both found to exhibit localized dynamics on the μs-ms timescale shown by several measures. First, many of the same residues exhibit R_2_-CPMG dispersions (Figure 7C,D); however, the associated rates of motions are different both between sites and between these homologues at the same sites. The most dramatic difference is position 164 that is quenched in hyrax BLVRB S164 but dynamic within human BLVRB T164. Second, comparisons of the extracted chemical shift changes from R_2_-CPMG dispersions (Δν_cpmg_) and the experimental CSPs lend insight into what the sampled conformations may be. For example, such comparisons were made to both bound NADPH and NADP^+^ for both apo human BLVRB (Appendix A) and apo hyrax BLVRB (Appendix A). Multiple residues are correlated to the experimental CSPs between apo and NADPH-bound forms, suggesting sampling of both enzymes to a bound conformation. Interestingly, several residues within the C-terminal lobe do not, suggesting that their inherent sampling differs. Third, mutational swaps at position 164, collected at 900 MHz (Figure 7E,F), further illustrate the localized nature of induced changes that form different networks (Figure 7G,H). Specifically, the R_2_-CPMG dispersions are impacted similarly by the mutational swaps for some residues (e.g., T110), some are unaffected by both swaps (e.g., H153), and some exhibit different responses (e.g., C109). Thus, even adjacent residues are impacted differently. However, a general finding for the mutational swaps is that, despite inducing the opposite changes to R_2_-CPMG dispersion at position 164, both mutants induce higher dispersion to distant residues or no effects.

Unlike our previous studies of holo human BLVRB [3], holo hyrax BLVRB maintains measurable R_2_-CPMG dispersion within key regions that include the active site (Figure 8A and Appendix A). This offers an opportunity to determine how the mutation at position 164 in hyrax BLVRB affects exchange prior to coenzyme release. In the holo form of hyrax BLVRB, the S164T mutation induces changes that can be observed to form a network from position 164 to the active site (Figure 8B,C and Appendix A). These imparted changes include residues adjacent to the S164T site along with residues 113–120 and 153–155 that are all proximal to the nicotinamide moiety. To determine the underlying cause of this increase in measured exchange for holo hyrax BLBRB observed in R_2_-CPMG dispersions, we also collected data at 600 MHz to quantify the associated biophysical parameters (Appendix A). The majority of holo hyrax BLVRB S164T changes are defined by extracted increases in the minor sampled populations (i.e., P_B_). The extracted chemical shift changes in holo hyrax BLVRB and the S164T mutant from R_2_-CPMG dispersions were once again compared to the experimental CSPs. As we only have experimental CSPs to the following step in the catalytic cycle, which is apo BLVRB, and not the ternary Michaelis–Menten complex (BLVRB/NADPH/substrate), we could plot these comparisons (Appendix A). Both the holo hyrax BLVRB and the S164T mutant do show correlations between these extracted and experimental CSPs, suggesting that some of the enzyme is dynamically sampling the apo conformation. Exceptions include several residues within the C-terminal lobe once again, as was found for apo BLVRB.

In summary, while coenzyme binding (i.e., the holo forms) induces quenching in both human and hyrax BLVRB homologues, hyrax BLVRB exhibits distinct dynamic behavior, particularly in the μs-ms timescale, as revealed by R_2_-CPMG dispersion data. While human BLVRB shows near-complete quenching of motion upon coenzyme binding, hyrax BLVRB retains measurable dynamics, especially in the active site that is likely due to the absence of the clamp. This retention of dynamics in holo hyrax BLVRB allowed for the identification of the induced changes upon mutation of the allosteric 164 site, indicating an increase in a sampled minor population.

## 3. Discussion

Here, we have extended studies from human BLVRB to hyrax BLVRB in order to address similarities and differences in function, primarily using solution NMR studies. These homologues comprise nearly 90% sequence identity, yet key distinctions include the absence of an active site arginine clamp in hyrax BLVRB and differences to a previously identified allosteric site at position 164. Probing these homologues here has provided mechanistic insight, suggesting that coenzyme isomerization is utilized by multiple BLVRB homologues.

Studies here have identified an important similarity between BLVRB homologues that reveal extensive reorganizations between bound forms of NADPH and NADP^+^, as recently reported for human BLVRB [10]. Our previous studies have shown that BLVRB-mediated NADPH-dependent reduction comprises multiple slow steps, yet an even slower rate-limiting step has remained unknown [9]. For example, studies of several homologues have shown that hydride transfer is 30–40 s^−1^, and coenzyme release is 2–10 s^−1^, yet catalytic turnover is just 0.1 s^−1^ [9]. As our studies here indicate that multiple BLVRB homologues undergo conformational changes between bound forms of NADPH and NADP^+^, this suggests that coenzyme isomerization might represent the rate-limiting step. This would be similar to other oxidoreductases such as aldose reductase [19] and glyceraldehyde-3-phosphate dehydrogenase [20]. In these other enzymes, conformational changes in the enzyme–coenzyme complex significantly impact catalytic efficiency, suggesting a conserved mechanism in redox-active enzymes that is likely at play for BLVRB family members as well.

Despite a shared large scale conformational change between human and hyrax BLVRB homologues, our studies indicate that these changes are not identical due to their different active sites. Namely, the hyrax BLVRB does not comprise an arginine clamp like most other family members. Considering that hyrax BLVRB has a slower turnover rate than human BLVRB, this would indicate that the evolutionary role of the active site clamp is to expedite this conformational reorganization and that hyrax BLVRB is slower due to the complete absence of a clamp.

Our studies also reveal that there are similarities and differences between the role of the allosteric 164 site in BLVRB homologues that evolutionarily toggles between a threonine and serine. For example, comparisons between BLVRB WT and position 164 mutational swaps here show that a threonine at this position in both homologues induces larger affinity differences between oxidative states of the coenzyme. Differences include distinctly perturbed networks in each homologue by mutational swaps that only affects the rate-limiting step in human BLVRB. Thus, while position 164 modulates both coenzyme binding and a rate-limiting step in human BLVRB, only coenzyme binding is modulated in hyrax BLVRB. Nonetheless, this suggests that the evolutionary substitution at position 164 plays a pivotal role in shaping the enzyme’s conformational landscape.

Our NMR relaxation experiments revealed inherent dynamics within both BLVRB homologues. Specifically, both homologues exhibit localized motions in similar regions on multiple timescales, yet their rates of conformational dynamics differ, as measured by R_2_-CPMG dispersion. While both homologues exhibit active site flexibility in the absence of coenzyme, hyrax BLVRB exhibits greater flexibility in its holo form due to the absence of the arginine clamp. Position 164 mutations induce largescale changes for both homologues that largely lead to increased exchange. Moreover, the finding that hyrax BLVRB exhibits measurable exchange even with the bound coenzyme allowed us to quantify the underlying dynamic basis of position 164 perturbations. Namely, mutation of hyrax BLVRB S164T induces larger transitions to a minor conformation, indicating once again that this site controls the dynamic landscape of the entire enzyme that includes the active site. Such perturbations to conformational sampling may underlie the functional control of coenzyme affinity mediated by this site in both BLVRB homologues.

Overall, our findings provide another example of a conformational change in the BLVRB-bound coenzyme after hydride transfer for multiple homologues and a deeper understanding of how evolutionary changes modulate function. These studies suggest that the unknown rate-limiting step is likely coenzyme isomerization. Furthermore, the differences observed between human and hyrax BLVRB highlight the critical role of allosteric regulation, particularly position 164, in determining catalytic efficiency and coenzyme discrimination.

## 4. Materials and Methods

### 4.1. Protein Expression and Purification

The expression and purification of recombinant human and hyrax BLVRB homologues were previously described [9]. Briefly, unlabeled proteins were expressed in LB media for 3 h at 37 °C, while labeled proteins were expressed in M9 minimal media for 5 h at 37 °C. For backbone assignments, ^2^H, ^15^N, and ^13^C-labeling was achieved by growing cultures in 99.9% D_2_O M9 minimal media supplemented with ^15^N-ammonium chloride and ^13^C-glucose. For R_2_-CPMG dispersion experiments, ^2^H,^15^N-labeling was performed using 99.9% D_2_O M9 minimal media supplemented with ^15^N-ammonium chloride. All BLVRB proteins were initially purified in a denatured state using Ni-affinity chromatography (Sigma-Aldrich, St. Louis, MO, USA), followed by refolding via dialysis. Further purification was conducted using Superdex 75 size exclusion chromatography (Cytiva, Marlborough, MA, USA), as previously described.

Mutations at position 164 were introduced using a nested PCR protocol. Initially, PCR was employed to generate a partial fragment containing the mutation, which was then used as a primer to produce the full-length PCR products. This product was subsequently inserted into an NdeI-cleaved pET21 vector. Hi-Fi PCR amplification (NEB, Ipswich, MA, USA) was performed, and the vector insertion was achieved using Takara’s In-Fusion technique (Takara Bio USA, Inc., San Jose, CA, USA).

### 4.2. Thermodynamics and UV-Kinetics Assays

For isothermal titration calorimetry (ITC) experiments, a MicroCal VP-ITC instrument (Malvern Panalytical, Northampton, MA, USA) was used to measure the binding interactions, with the enzyme concentration maintained at 100 nM and the ligand (either NADPH or NADP^+^) at 800 µM. The buffers used for ITC experiments were identical to those utilized for NMR sample preparation. The measurements were conducted in triplicate at 20 °C, and data were processed using NITPIC and Sedphat. Reported values represent the mean and standard deviation from these replicates.

For steady-state UV kinetic assays, a SpectraMax Plus reader was employed to monitor NADPH oxidation to NADP^+^ at 340 nm, using a total reaction (Molecular Devices, San Jose, CA, USA) volume of 300 µL and a 1 mm path length. These reduced volumes were chosen to minimize path length and prevent signal saturation. The NADPH concentration was held constant at 400 µM; the BLVRB enzyme was held constant at 20 µM, while varying concentrations of the FAD substrate were tested. Reaction velocities were determined from changes in absorbance during the initial minute of the reaction, with triplicate kinetic measurements averaged. Data were fitted to the Michaelis–Menten equation using GraphPad Prism software Version 10.4.0 (GraphPad Software Inc., San Diego, CA, USA) to calculate kinetic parameters. Velocity corrections to µM/s were calculated using the extinction coefficient of NADPH (6222 M^−1^ cm^−1^).

### 4.3. NMR Sample Preparation, Spectroscopy, and Data Analysis

All BLVRB samples were prepared in 50 mM bis-tris (pH 6.5), 50 mM NaCl, and 1 mM DTT, with a final concentration of 500 µM and 8% D_2_O. Ternary complexes of BLVRB with NADP^+^ and BR were prepared with 1 mM coenzyme and 700 µM BR. Backbone assignments for hyrax BLVRB, using ^2^H, ^15^N, ^13^C-labeled enzymes, were obtained on a Bruker 600 MHz spectrometer (Billerica, MA, USA) equipped with a cryo-probe at 20 °C. This included HNCA, HN(co)CA, HNCACB, and HN(co)CACB spectra. All 3D assignment spectra were acquired using non-uniform sampling (NUS), reconstructed to 72 and 96 increments in the nitrogen and carbon dimensions, respectively. Backbone assignments have been deposited for apo hyrax BLVRB, holo hyrax BLVRB with NADP^+^, and holo hyrax BLVRB with NADPH in the Biological Magnetic Resonance Data Bank (BMRB), with accession numbers 52642, 52643, and 52644, respectively.

R_2_-CPMG relaxation experiments were conducted on both the Rocky Mountain Varian 900 MHz spectrometer equipped with a cryo-probe and the Bruker 600 MHz spectrometer at 20 °C using TROSY selection. Data were analyzed to extract exchange rates and populations using GUARDD [21].

Amide CSPs were calculated as the square root of the sum of squares of the differences in nitrogen and proton chemical shifts between two states. All spectra were processed and analyzed with NMRPipe [22] and CCPNmr [23].

For supplementary ^13^C-filtered NOESY experiments on the human BLVRB bound to coenzyme, data were collected via the Rocky Mountain Varian 900 MHz spectrometer. To detect ligand signals within the protein complex, a lower concentration of 450 μM ligand was used than 500 μM human BLVRB in 100% D_2_O. To assign the resonances in the NOE spectrum of NADPH and NADP^+^, we referenced the known chemical shift values and NOE patterns of the free forms of the coenzymes available in the BMRB database. Additionally, the NOE spectrum provides cross-peaks that indicate spatial proximity between nuclei. For instance, H6 is closer to the ribose protons, resulting in stronger intensities and more pronounced correlations than H5.

### 4.4. Bilirubin Purification, and Analysis

Due to the unavailability of BR β in the market, β was isolated from an isomer mixture (Sigma-Aldrich, St. Louis, MO, USA). Initially, BR was dissolved in 0.1 M NaOH and then separated using an Agilent C8 column via HPLC. The mobile phases used were water with 0.1% FA (buffer A) and acetonitrile with 0.1% FA (buffer B). β, γ, and δ BR were eluted when increasing the B percentage from 10 to 20%, followed by the elution of α. The quality of the separated BR and the identification of the isomer were determined by dissolving it in CDCl_3_ and performing a ^1^H NMR analysis.

## Figures and Tables

**Figure 1 ijms-25-13233-f001:**
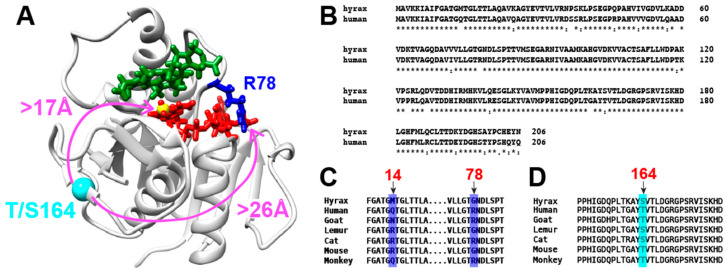
The structure and sequence differences between hyrax and human BLVRB homologues. (**A**) X-ray crystal structure of a holo BLVRB illustrating the coenzyme and substrate binding sites (PDB accession 1HE3). The coenzyme is NADP^+^ (red), and the substrate binding site is illustrated with bound mesobiliverdin IV (green). Also shown are the active site R78 clamp (blue) and the distal allo-steric site (cyan). (**B**) Full-sequence comparisons between hyrax and human BLVRB homologues. (**C**) Sequence comparisons of the evolutionary variability of BLVRB positions 14 and 78 with the active site (blue). (**D**) Sequence comparisons of the allosteric BLVRB position 164 that evolutionarily swaps between either a threonine or serine (cyan). “*” is for conserved, “.” partially conserved.

**Figure 2 ijms-25-13233-f002:**
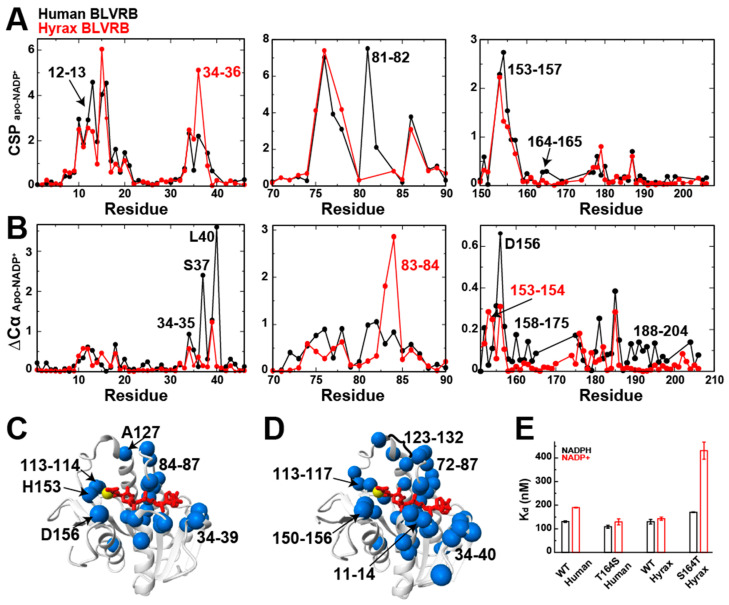
Hyrax BLVRB Cα changes between apo and holo forms and comparison to the human BLVRB homologue. (**A**) Amide CSPs between apo and NADP^+^-bound forms of human BLVRB (black) and hyrax BLVRB (red). (**B**) Cα chemical shift differences (∆Cα) between apo and NADP^+^-bound forms of human BLVRB (black) and hyrax BLVRB (red). (**C**) Hyrax BLVRB Cα differences larger than the average plus ½ standard deviation (0.27 ppm) mapped onto the X-ray crystal structure of human BLVRB. Blue spheres represent imparted changes, red is the coenzyme, and the yellow sphere is the hydride site. (**D**) Human BLVRB Cα differences larger than 0.27 ppm mapped onto the X-ray crystal structure of human BLVRB. (**E**) Binding affinities for NADPH (black) and NADP^+^ (red) in human and hyrax BLVRB, including the WT for each, human BLVRB T164S, and hyrax BLVRB S164T mutants. All data were collected at 900 MHz at 20 °C.

**Figure 3 ijms-25-13233-f003:**
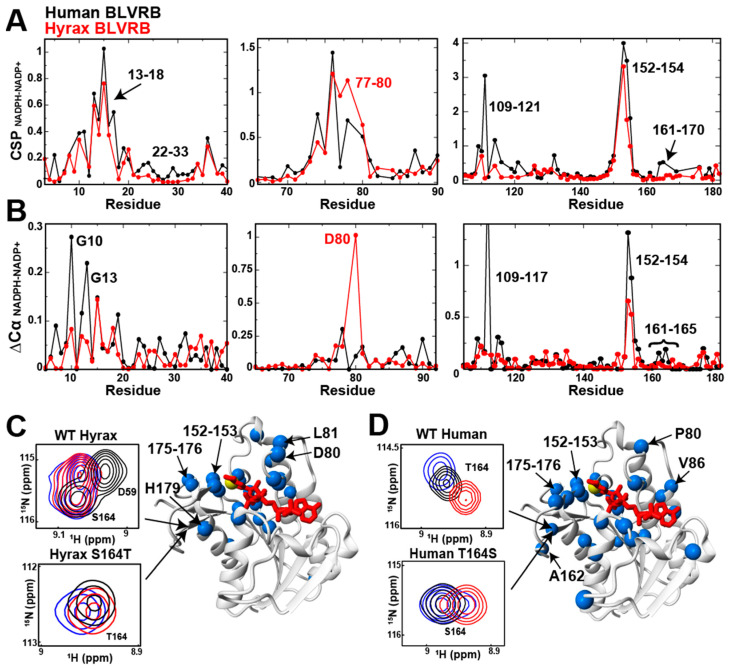
Hydride-induced changes between human and hyrax BLVRB homologues. (**A**) CSPs between NADPH- and NADP^+^-bound forms of human BLVRB (black) and hyrax BLVRB (red). Key residues showing significant CSPs are labeled, including the regions 13–18, 77–80, and 161–170. (**B**) Differences in Cα chemical shifts (ΔCα) between NADPH- and NADP^+^-bound states in human BLVRB (black) and hyrax BLVRB (red) are shown for key regions of the proteins. Residues with greater ΔCα values in hyrax BLVRB are labeled in red, and those with greater changes in human BLVRB are labeled in black. (**C**) Hyrax BLVRB Cα chemical shift changes (ΔCα > 0.1 ppm) mapped onto the X-ray crystal structure of human BLVRB (blue spheres) between bound NADPH and NADP^+^ forms. The coenzyme’s hydride position is also shown (yellow sphere). The left insets show 15N-HSQC spectra for WT hyrax BLVRB (top) at position S164 and the S164T mutant (bottom) across apo, NADPH-bound, and NADP^+^-bound forms. (**D**) Human BLVRB Cα chemical shift changes (ΔCα > 0.1 ppm) mapped onto the X-ray crystal structure of human BLVRB (blue spheres) between bound NADPH and NADP^+^ forms. The coenzyme NADPH is shown as a yellow sphere. The left inset shows the 15N-HSQC spectra of both human WT and T164S BLVRB at position T164 for the apo (black), NADPH-bound (blue), and NADP^+^-bound (red) states. All data were collected at 900 MHz at 20 °C.

**Figure 4 ijms-25-13233-f004:**
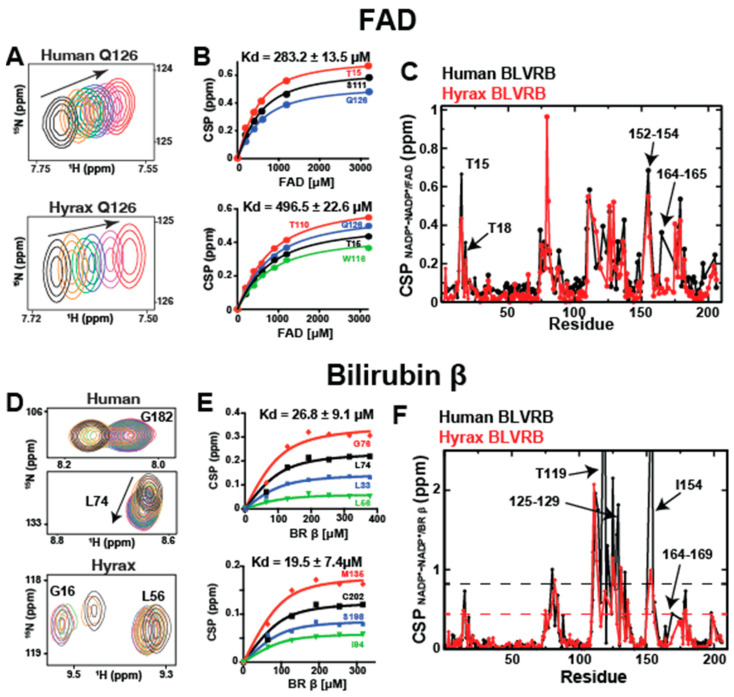
FAD substrate and bilirubin-β product binding to human and hyrax BLVRB. (**A**) 15N-HSQC spectra showing the CSPs for residue Q126 in holo forms of human BLVRB (top) and hyrax BLVRB (bottom) upon titrations with FAD. (**B**) Binding isotherms for FAD titrations for human BLVRB (top) and hyrax BLVRB (bottom). The calculated binding affinity values are indicated for each homologue. (**C**) Global FAD-induced CSPs for human BLVRB (black) and hyrax BLVRB (red). (**D**) ^15^N-HSQC spectra illustrating CSP examples of both slow and fast exchange upon titrations of holo forms of human BLVRB (top) and hyrax BLVRB (bottom) with BR-β. (**E**) Binding isotherms for the subset of residues under the fast-exchange binding regime for BR-β titrations for human BLVRB (top) and hyrax BLVRB (bottom). The calculated binding affinity values are indicated for each homologue. (**F**) Global BR-β induced CSPs for human BLVRB (black) and hyrax BLVRB (red). All data were collected at 900 MHz at 20 °C.

**Figure 5 ijms-25-13233-f005:**
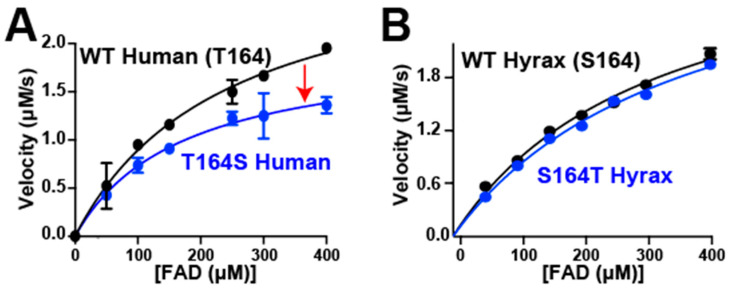
Kinetic analysis and binding affinities for BLVRBs. (**A**) Initial velocities of FAD concentration for WT human BLVRB (black) and the T164S mutant (blue) measured via UV kinetics. The reactions were conducted with saturating NADPH and varying FAD concentrations, monitored at 340 nm and fit to the Michaelis–Menten model. The T164S mutant shows reduced catalytic efficiency compared to the WT, as evidenced by lower velocities at higher FAD concentrations. (**B**) Initial velocities of FAD concentration for WT hyrax BLVRB (black) and the S164T hyrax mutant (blue) measured under the same conditions as (**A**). In contrast to the human BLVRB, the S164T mutation in hyrax BLVRB does not significantly alter catalytic efficiency.

**Figure 6 ijms-25-13233-f006:**
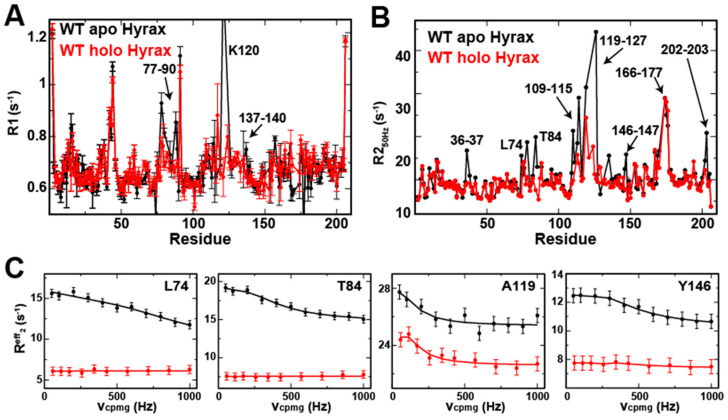
Relaxation rates for both apo and holo hyrax BLVRB. (**A**) R1 relaxation rates are shown for apo (black) and holo (red, NADP^+^ bound). (**B**) R_2_ relaxation rates extracted from R_2_-CPMG dispersions at 50 Hz refocusing field are shown for apo (black) and holo (red, NADP^+^ bound). (**C**) Specific examples of full R_2_-CPMG dispersions for both apo hyrax (black) and holo hyrax (red). All data were collected at 900 MHz at 20 °C.

**Figure 7 ijms-25-13233-f007:**
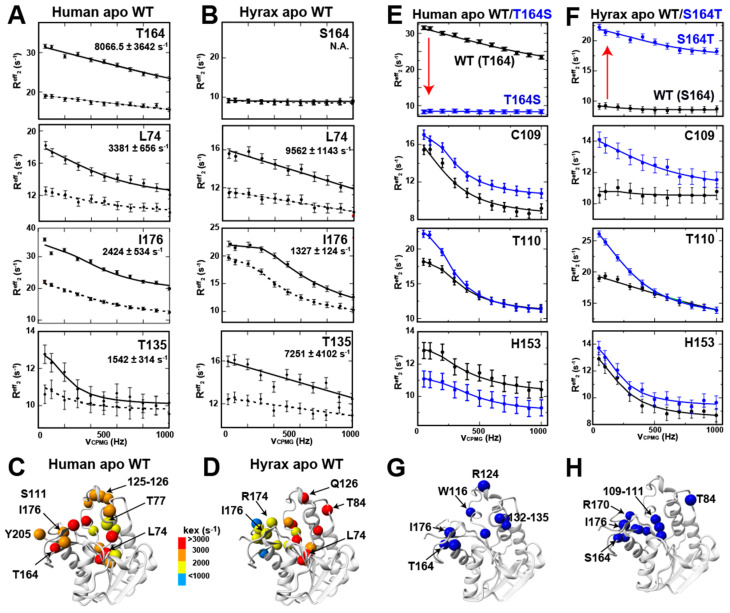
R_2_-CPMG dispersion for apo human and apo hyrax BLVRB. (**A**) Human BLVRB R_2_-CPMG dispersions are shown simultaneously fit at 900 MHz (solid lines) and 600 MHz (dashed lines), which were previously reported [3]. (**B**) Hyrax BLVRB R_2_-CPMG dispersions are shown simultaneously fit at 900 MHz (solid lines) and 600 MHz (dashed lines). (**C**) Human BLVRB exchange rates are mapped onto the human BLVRB colored (spheres) according to their fit exchange rates. (**D**) Hyrax BLVRB exchange rates are mapped onto the human BLVRB colored (spheres) according to their fit exchange rates. (**E**) R_2_-CPMG dispersion at 900 MHz for human WT apo BLVRB (black) and human T164S apo BLVRB (blue). (**F**) R_2_-CPMG dispersion at 900 MHz for hyrax WT apo BLVRB (black) and hyrax T164S apo BLVRB (blue). (**G**) The residues of human apo BLVRB showing significant changes upon mutation are mapped onto the structure of human BLVRB. (**H**) The residues of hyrax apo BLVRB showing significant changes upon mutation are mapped onto the structure of human BLVRB. All data were collected at 900 MHz at 20 °C.

**Figure 8 ijms-25-13233-f008:**
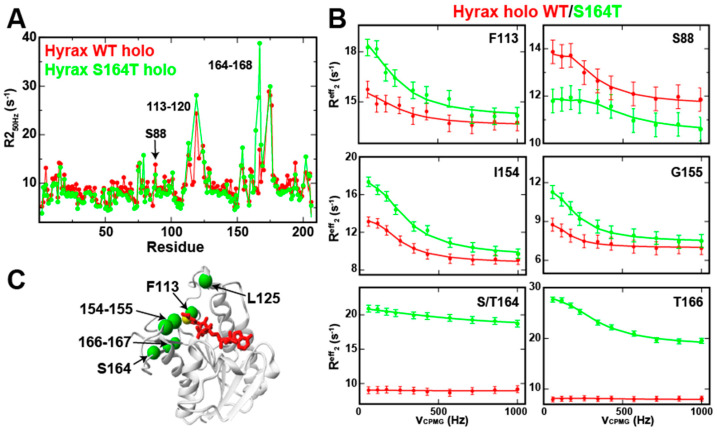
R_2_-CPMG dispersion for holo hyrax BLVRB WT versus the S164T mutation. (**A**) R_2_ relaxation rates extracted from R_2_-CPMG dispersions at 50 Hz refocusing field are shown for holo hyrax BLVRB WT (red, NADP^+^ bound) and hyrax BLVRB S164T (green, NADP^+^ bound). (**B**) R_2_-CPMG dispersions are shown at 900 MHz for both holo hyrax BLVRB WT and the S164T mutation fit at both 900 MHz and 600 MHz simultaneously. All data were collected at 900 MHz at 20 °C. (**C**) Residues exhibiting changes between holo hyrax BLVRB WT and the S164T mutation are mapped onto the X-ray crystal structure of human BLVRB (green spheres).

**Table 1 ijms-25-13233-t001:** Thermodynamics ^a^ and kinetics ^b^ parameters.

BLVRB	K_d_ (nM) with NADPH	K_d_ (nM) with NADP^+^	ΔG (kcal/Mol) with NADPH	ΔH (kcal/Mol) with NADPH	ΔS (kcal/Mol) with NADPH	ΔG (kcal/Mol) with NADP^+^	ΔH (kcal/Mol) with NADP^+^	ΔS (kcal/Mol) with NADP^+^	ΔG (kcal/Mol) with NADP^+^	k_cat_ (s^−1^)	K_M_ (μM) FAD
Human ^c^	130 ± 10	190 ± 5	−9.23 ± 0.02	−11.8 ± 0.7	−8.9 ± 2.4	−9.01 ± 0.00	−9.8 ± 0.5	−2.7 ± 1.9	−9.01 ± 0.00	0.16 ± 0.03	242 ± 36
Human T164S ^c^	110 ± 10	130 ± 10	−9.34 ± 0.03	−8.5 ± 0.5	1.2 ± 0.1	−9.25 ± 0.11	−9.1 ± 0.2	0.5 ± 1	−9.25 ± 0.11	0.10 ± 0.03	171 ± 33
Hyrax	130 ± 10	140 ± 5	−9.24 ± 0.04	−9.6 ± 0.3	−1.3 ± 0.8	−9.18 ± 0.04	−10.5 ± 0.5	−4.7 ± 1.9	−9.18 ± 0.04	0.11 ± 0.01	184 ± 24
Hyrax S164T	170 ± 2	430 ± 40	−9.08 ± 0.02	−12.9 ± 0.6	−12.3 ± 0.9	−8.54 ± 0.11	−9.3 ± 0.8	−3.0 ± 3.2	−8.54 ± 0.11	0.09 ± 0.01	233 ± 70

^a^ Thermodynamic parameters were measured via ITC. ^b^ Kinetics parameters were measured using UV kinetics to monitor the oxidation of NADPH. ^c^ NADP^+^ thermodynamic data were similar to those previously published [14].

## Data Availability

The original contributions presented in this study are included in the article/Appendix A. Further inquiries can be directed to the corresponding author(s).

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
