# Peer review of "Evolutionary Adaptations in Biliverdin Reductase B: Insights into Coenzyme Dynamics and Catalytic Efficiency"

_ijms, 2024, doi:10.3390/ijms252413233_

Round 1
Reviewer 1 Report
Comments and Suggestions for Authors
The article by Lee and co-workers reports the structural and functional differences between human BLVRB and its hyrax homologue, focusing on evolutionary adaptations at the active site and allosteric regions.
According to my opinion, this paper will be of medium interests for the readers of the International Journal of Molecular Sciences and needs slight modifications before being accepted for publication.
The authors should provide more details on ITC experiments. The raw data of ITC titrations and the resulting curve fitted should be reported at least in Supporting Information. Were deltaH and deltaS comparable between all the analyzed systems?
Some minor points:
- The caption of Fig 2E is missing.
- The caption of Figure 1 should be amended. There is a typo in fig 1A) “and ?? (green)”. A wrong caption has been reported for figure 1C.
- The authors should introduce all the acronyms, including FAD, FMN, NADPH..
- Some typos are present, i.e. pag 5 line 181 “”state vs states”
In view of the above, according to my opinion, the paper is suitable for publication in IJMS after minor revisions.
Author Response
Reviewer: The authors should provide more details on ITC experiments. The raw data of ITC titrations and the resulting curve fitted should be reported at least in Supporting Information. Were deltaH and deltaS comparable between all the analyzed systems?
Response: We apologize for not providing more of the data and have corrected this. We have now included all of the raw data as Supplementary Figure 2 (and moved all the other Supplementary Figures accordingly). We have also added in Table 1 the deltaH and deltaS terms, but the deltaH terms are highly similar with the deltaS terms largely dictating the binding differences.
Some minor points:
Reviewer: The caption of Fig 2E is missing.
Response: We apologize for not having the full caption and we have explicitly completed this now. We thank the reviewer for pointing this out.
Reviewer: The caption of Figure 1 should be amended. There is a typo in fig 1A) “and ?? (green)”. A wrong caption has been reported for figure 1C.
Response: We completely agree and thank the reviewer for pointing this out. We have now specifically stated the PDB accession number as well and have provided the complete information for the structural information provided.
Reviewer: The authors should introduce all the acronyms, including FAD, FMN, NADPH.
Response: We have now explicitly defined these the first time that they are mentioned and we thank the reviewer for this importance suggestion.
Reviewer 2 Report
Comments and Suggestions for Authors
The authors have studied human and hyrax BLVRB protein in its apo form as well as with bound NADPH or NADP+. In addition, complexation with substrate and product was studied. Single-point mutants were also included in the study. The authors used advanced NMR techniques to describe the structural and dynamic differences and provide a lot of experimental results with appropriate analyses.
In my opinion, the manuscript deserves publication in IJMS. Overall, the work is adequately designed and the results are well described. During reading the manuscript, I came across several unclear points:
1. Caption to Fig. 1 contains "??" and (C) is not the allosteric 164
2. I think it might clarify the workflow if you mention the mutations T164S/S164T earlier in the article, e.g. in the last paragraph of Introduction. Now it first appears on l. 164 (line, not residue, what a coincidence).
3. Regarding the mutations, have the authors tried to change other residues? For instance, R78G mutations of the human protein and G78R of hyrax would, in my view, bring new insights into the role of the clamp.
4. line 142: although residues 34-40 are not much affected in Cα shifts as stated in the text, there are large CSPs in NH, this should be explained
5. Caption to Fig. 2 should be revised (e.g., no lemur in this work; Ca instead of Cα; unclear if 0.27 ppm refers to std or its sum with mean; duplicate numbering of panels)
6. l. 186-188: incomplete sentence?
7. Fig. 3: since structure of the figure is similar to Fig. 2, consider moving some explanatory parts of the caption to the caption of Fig. 2
8. On l. 212-218, the effects of 164 mutations on NH shift changes after NADPH/NADP+ binding are discussed. Hyrax S164T is claimed to cause allosteric coupling. However, the peak in Fig. 3C seems not to change. On the other hand, human T164S in Fig. 3D shifts more. This needs a better explanation. (Or are the panel labels in Fig. 3C and D incorrect? Please check the caption of Fig. 3 in this respect as well.)
9. L. 249-271: The authors have studied FAD and BR-β binding. I don't understand, why not a proper substrate-product pair. I was expecting comparison of biliverdin vs. bilirubin binding, or FAD vs. FADH2 binding. Could you please comment on this?
10. The graphics Fig. 4 has rather low quality, please check this (and other figures) in future revisions
11. Caption to Fig. 4 D doesn't match the assignments in the spectra shown
12. L. 284-286: "similarly" in which sense? Dissociation constants are quite different, and the kinetic aspects have not been discussed so far. I don't see how the statement on l. 284-286 is supported by the data.
13. L. 296 typo nM instead of nm
14. L. 300: the rates of human T164S and hyrax WT do not look the same in Fig. 5. Could you please check the underlying data or explain better? Matching the vertical scales of Fig. 5A and B to each other would also help the reader.
15. Tab. 1: typo in the caption, missing a,b,c in the footnotes
16. Fig. 6 B: the labels on the vertical axis do not match the ticks. In addition, turning the tick direction outwards would make the plot clearer.
17. L. 370: Should it be Fig. S4 instead of Fig. S3?
18. Fig. 7:
a. the uncertainties of the rates should be rounded to one or two digits precision and the values correspondingly (tables in SI are much better in this aspect)
b. I would expect CMPG dispersion data for some of the interesting residues labeled in Fig. 6B, e.g. some of 119-127, to appear in Fig. 7
c. in panel F, the values of WT S164 rates do not match the R2's presented in Fig. 6 B. May it be due to the axis problem noted in point 16 above?
d. last sentence of the caption is not valid here
19. Scheme 2 (i.e., SI, Fig. 2):
a. the presented numbering of base atoms is not standard, e.g., adenine "H5" is in fact H8;
b. could you please explain how you assigned the NADPH and NADP+ proton resonances in the complexes?
Author Response
Reviewer: 1. Caption to Fig. 1 contains "??" and (C) is not the allosteric 164
Response: We thank the reviewer for pointing out these errors and we have corrected them along with fixing (C) that detailed the active site 14/78 evolutionary changes.
Reviewer: 2. I think it might clarify the workflow if you mention the mutations T164S/S164T earlier in the article, e.g. in the last paragraph of Introduction. Now it first appears on l. 164 (line, not residue, what a coincidence).
Response: We completely agree with the reviewer and have placed the first description of these swapped mutants a the end of the Introduction (specifically, the second to last line of the Introduction when we describe the functional changes due to position 164 swaps).
Reviewer: 3. Regarding the mutations, have the authors tried to change other residues? For instance, R78G mutations of the human protein and G78R of hyrax would, in my view, bring new insights into the role of the clamp.
Response: This is an important point brought up by the reviewer, which we have now tried to clarify within the section entitled, “The oxidative state of the coenzyme induces largescale changes for both homologues that suggests coenzyme isomerization.” Previously, active site mutations were made in the context of human BLVRB that also include hyrax BLVRB (see Duff et al, Biochemistry Journal 2020). These active site mutations did indeed modulate coenzyme release, but these changes were not correlated to the rate-limiting step. That is why our studies here that identify largescale changes upon hydride transfer (i.e., the oxidative state of the coenzyme) strongly suggest that coenzyme isomerization is the rate-limiting step. We have specifically stated this in this section, as it is an important outcome of this study and apologize for any lack of clarity.
Reviewer: 4. line 142: although residues 34-40 are not much affected in Cα shifts as stated in the text, there are large CSPs in NH, this should be explained
Response: We apologize for any lack of clarity, as similar regions do exhibit changes upon coenzyme binding. We have changed this section to reflect that both homologues do exhibit changes, but that these are slightly smaller in hyrax BLVRB.
Reviewer: 5. Caption to Fig. 2 should be revised (e.g., no lemur in this work; Ca instead of Cα; unclear if 0.27 ppm refers to std or its sum with mean; duplicate numbering of panels)
Response: We apologize for this error and have corrected the figure legend.
Reviewer: 6. l. 186-188: incomplete sentence?
Response: We have rewritten this section based on your important point in (3) and apologize for any errors.
Reviewer: 7. Fig. 3: since structure of the figure is similar to Fig. 2, consider moving some explanatory parts of the caption to the caption of Fig. 2
Response: It is true that Figure 3, which focuses on NADPH/NADP+ changes along with resonances positions of apo, is somewhat similar to Figure 2, which focuses on apo/holo changes. We apologize, but we are unsure of what specific parts of the structural caption that the reviewer would change.
Reviewer: 8. On l. 212-218, the effects of 164 mutations on NH shift changes after NADPH/NADP+ binding are discussed. Hyrax S164T is claimed to cause allosteric coupling. However, the peak in Fig. 3C seems not to change. On the other hand, human T164S in Fig. 3D shifts more. This needs a better explanation. (Or are the panel labels in Fig. 3C and D incorrect? Please check the caption of Fig. 3 in this respect as well.)
Response: We apologize, as the legends for Figure 3C and 3D were switched that may help clarify our findings better than before. As the reviewer stated and is stated within the main text, “Hyrax BLVRB showed no obvious coupling between these forms at position S164.” Only when position 164 comprises a threonine are there any CSPs observed upon coenzyme binding. We apologize for any lack of clarity that our Figure legend error may have caused.
Reviewer: 9. L. 249-271: The authors have studied FAD and BR-β binding. I don't understand, why not a proper substrate-product pair. I was expecting comparison of biliverdin vs. bilirubin binding, or FAD vs. FADH2 binding. Could you please comment on this?
Response: This is a good question that we apologize for not clarifying further. FADH2 is highly unstable, so we opted for purifying the bilirubin-beta product. Unfortunately, the bilirubin-beta substrate was unsuccessfully purified.
Reviewer: 10. The graphics Fig. 4 has rather low quality, please check this (and other figures) in future revisions
Response: We have increased this resolution and apologize for the initial poor resolution.
Reviewer: 11. Caption to Fig. 4 D doesn't match the assignments in the spectra shown 2
Response: We apologize for neglecting to update this figure legend, which has now been corrected.
Reviewer: 12. L. 284-286: "similarly" in which sense? Dissociation constants are quite different, and the kinetic aspects have not been discussed so far. I don't see how the statement on l. 284-286 is supported by the data.
Response: We apologize for any lack of clarity and have refined this statement to be more explicit. Our intention was to indicate that the FAD substrate binds weakly to both homologues and that the BR-beta product is the same within uncertainty.
Reviewer: 13. L. 296 typo nM instead of nm
Response: We have fixed this typo and appreciate the Reviewer pointing this out.
Reviewer: 14. L. 300: the rates of human T164S and hyrax WT do not look the same in Fig. 5. Could you please check the underlying data or explain better? Matching the vertical scales of Fig. 5A and B to each other would also help the reader.
Response: We have rechecked the Vmax values that were nearly identical for human T164S and hyrax WT, which give similar kcat values when corrected for concentration.
Reviewer: 15. Tab. 1: typo in the caption, missing a,b,c in the footnotes
Response: We apologize and have included these footnotes.
Reviewer: 16. Fig. 6 B: the labels on the vertical axis do not match the ticks. In addition, turning the tick direction outwards would make the plot clearer.
Response: We have corrected both the tick direction and matched the Y-axis correctly. We apologize for this error and thank the Reviewer for pointing this out.
Reviewer: 17. L. 370: Should it be Fig. S4 instead of Fig. S3?
Response: The Reviewer is absolutely correct and these have been changed. We apologize for this error.
Reviewer: 18. Fig. 7: a. the uncertainties of the rates should be rounded to one or two digits precision and the values correspondingly (tables in SI are much better in this aspect).
Response: We apologize for missing this and have corrected this mistake.
Reviewer: 18b. I would expect CMPG dispersion data for some of the interesting residues labeled in Fig. 6B, e.g. some of 119-127, to appear in Fig. 7
Response: We apologize for not including full R2-CPMG dispersions of some of these examples. As Figure 7 includes full R2-CPMG dispersions fit at both 900 and 600 MHz to illustrate local differences, we have placed these full R2-CPMG dispersions into Figure 6C and thank the Reviewer for his/her suggestion.
Reviewer: c. in panel F, the values of WT S164 rates do not match the R2's presented in Fig. 6 B. May it be due to the axis problem noted in point 16 above?
Response: The Reviewer is correct in thinking that the R2eff of Hyrax BLVRB S164 appears higher in Figure 6b than it does in Figure 7b, but the elevated R2eff values only include residues 166-177. Thus, to clarify further, we have specifically labeled residues 166-177 within Figure 6b, as these do not include Hyrax S164 that has a relatively low R2eff~10 Hz.
Reviewer: d. last sentence of the caption is not valid here
Response:We believe that we have corrected this and apologize for any error.
Reviewer: 19. Scheme 2 (i.e., SI, Fig. 2): a. the presented numbering of base atoms is not standard, e.g., adenine "H5" is in fact H8;
Response: We have corrected this and apologize, as we had used the nomenclature listed within the BMRB.
Reviewer: b. could you please explain how you assigned the NADPH and NADP+ proton resonances in the complexes?
Response:We apologize for not including this description, which we have not placed within the Methods section.
Reviewer 3 Report
Comments and Suggestions for Authors
Please see the attached file.

Author Response
Reviewer: 1. The resolution of Scheme 2 is bad. Please consider enhancing the resolution.
Response: We apologize for any low-resolution figure and have enhanced the resolution of Supplementary Figure 2.
Reviewer: 2. The Introduction does not provide a comprehensive background. A detailed
introduction about this field should be included. Also, the significance of the
authors’ work, what have the authors done in the present work, and the motivation
of this work should be covered in the Introduction.
Response: We appreciate the Reviewer’s suggesting and agree that there should be a more expansive summary of previous work. Thus, we have now included a more extensive background in the Introduction, expanding the second paragraph to now three paragraphs that include our collaborative studies over the last 6 years along with the first X-ray crystal structure, which is referenced now. Thus, for clarity, the first paragraph introduces the biological importance and general functions of BLVRB while the second through fourth paragraphs now comprises a summary of the biochemical studies and our motivation to study Hyrax BLVRB that does not comprise an active site clamp.
Reviewer: 3. Except for the NMR, are there any other spectroscopy methods that can be used to
detect such as coenzyme binding, molecular interactions? Please discuss this in
the manuscript.
Response: This is a very good question that we completely agree should be addressed. In fact, we thought it so important that we have placed several sentences within the Introduction (third paragraph) discussing specifically the advantage of utilizing NMR solution experiments to identify the conformational changes with appropriate citations.
Reviewer: 4. What contributions will this manuscript bring to the community? Please provide the
details.
Response: We have significantly changed the Introduction and our final discussion to try to emphasize our contributions. Namely, the findings of conformational changes after hydride transfer indicate coenzyme isomerization, which is likely the rate-limiting step. Furthermore, key differences between homologues within both their active sites and allosteric regulation via position 164 are also highlighted, which likely contributes to differences in the rates of isomerization.